# Study of the Influence of the Dielectrophoretic Force on the Preferential Growth of Bacterial Biofilms in 3D Printed Microfluidic Devices

Alexandra Csapai [1,*], Dan Alexandru Toc [2,*], Violeta Pascalau [1], Nicoleta Tosa [3], Septimiu Tripon [4], Alexandra Ciorîță [4], Razvan Marian Mihaila [5], Bogdan Mociran [6], Carmen Costache [2] and Catalin O. Popa [1]

1   Materials Engineering Department, Technical University of Cluj-Napoca, 103-105 Muncii Ave., 400641 Cluj-Napoca, Romania
2   Department of Microbiology, Iuliu Hatieganu University of Medicine and Pharmacy, 8 Victor Babes, Street, 400012 Cluj-Napoca, Romania
3   National Institute for Research and Development of Isotopic and Molecular Technologies, Molecular and Biomolecular Physics Department, 67-103 Donat Street, 400293 Cluj-Napoca, Romania
4   Integrated Electron Microscopy Laboratory, National Institute for Research and Development of Isotopic and Molecular Technologies, 67-103 Donat Street, 400293 Cluj-Napoca, Romania
5   Department of Ophthalmology, Centre Hospitalier Régional d'Orléans, 14 Av. de l'Hôpital, 45100 Orléans, France
6   Faculty of Electrical Engineering, Technical University of Cluj-Napoca, 26-28 G. Barițiu Street, 400394 Cluj-Napoca, Romania
*   Correspondence: alexandra.csapai@stm.utcluj.ro (A.C.); toc.dan.alexandru@elearn.umfcluj.ro (D.A.T.)

**Abstract:** Understanding the effect of different electric potentials upon the preferential formation of biofilms inside microfluidic devices could represent a step forward in comprehending the mechanisms that govern biofilm formation and growth. 3D printed microfluidic devices were used to investigate the influence of the dielectrophoretic forces on the formation and growth of *Staphylococcus aureus* ATCC 25923 biofilms. Bacterial suspensions of 2.5 McF were pushed through microfluidic channels while simultaneously applying various potential differences between 10 and 60 V. The overall electric field distribution within the channel was simulated using the COMOSL software. The effect of the electric potential variation on the preferential biofilm formation was determined using an adjusted microtiter plate technique, as well as a qualitative method, Scanning Electron Microscopy (SEM). SEM images were used to describe the morphology of the biofilm surface. The conclusions show that the dielectrophoretic forces, resulting due to inhomogeneity of the electric field, have more visible effects upon the cells up to 40 V. Above this magnitude, due to a more homogenous distribution of the electric field, the formation and growth of the biofilm become more uniform. At around 60 V, the distance between the high electric gradient regions decreases, leading to an almost uniform distribution of the electric field and, therefore, to a shift from dielectrophoretic to electrophoretic forces acting upon the bacterial cells.

**Keywords:** microfluidic devices; biofilms; dielectrophoresis; additive manufacturing

## 1. Introduction

Numerous research areas, such as regenerative medicine, tissue engineering, biotechnology, drug discovery, and cancer research, treatment, and diagnostics, rely on fundamental techniques of cell separation, sorting, and analysis. One method that has arisen interest in the last few decades within the variety of sorting techniques is the microfluidic-based cell separation method [1]. Microfluidic devices allow the "handling of fluids in technical apparatus having internal dimensions in the range of micrometres up to a few millimetres". Some of the main advantages of microfluidic devices are the existence of a laminar flow within the system, the possibility of direct interactions with cells, faster and

parallel sample analysis, increased surface-to-volume ratio, as well as the possibility of device automation [2].

Among the most known microfluidic devices is the H-type filter, which takes advantage of the diffusive mixing between the adjacent laminar streams to facilitate the passive separation of particles. By means of controlling parameters such as the channel geometry and flow rate, the diffusion time of the particles within the flow can be limited, therefore achieving cell isolation, manipulation, and separation [3]. These properties of the microfluidic devices, together with their ability to provide unique control over the flow conditions, high throughput, and the capacity to emulate in vivo-like biological environments, make them promising platforms for bacterial biofilm observation and research [4].

A novel method in the fabrication of microfluidic systems is additive manufacturing (AM). AM for microfluidic devices is attracting widespread interest [5] due to its revolutionary approach of creating parts or prototypes layer-by-layer directly from a computer-aided design (CAD). One AM method that stands out is fused deposition modelling (FDM), which allows the use of numerous biocompatible thermoplastic polymers for the fabrication of different printed parts. Some of the most recent works employ the 3D printing technique for the fabrication of flexible thermoplastic polyurethane microfluidic devices [6], high-pressure, heat-resistant, transparent PLA devices [7], and even 3D printed microfluidic devices with integrated materials such as wires, glass and electrodes [5,8]. Moreover, 3D printing allows for the production of customizable devices, with structural components such as porosity, pore–to–pore distance, geometry, and surface roughness specially tailored for specific applications. From generating forebrain-specific organoids from human induced pluripotent stem cells (iPSCs) [9] through miniaturized, modular spinning bioreactors, to high-throughput fabrication of hydrogel scaffold droplets for cell encapsulation [10], and directly immobilizing and maintaining the viability and functionality of 3D multicellular spheroids [11], 3D printed microfluidic devices have proven to be an unparalleled and vital platform for studying single cells, large cell populations, cell interactions, and organoids [12]. Therefore, for this research, the FDM 3D printing method was chosen.

Throughout the years, various particle manipulation techniques within microfluidic devices were developed, including methods based on properties such as:

- optical properties [13,14] for cell sorting, trapping, single-cell analysis, molecule injection, and electroporation;
- magnetic properties [15] for cell separation, capturing, trapping, isolation, and analysis;
- electric properties [16–18] for cell separation, focusing, isolation, and cell fusion;
- mechanical properties [19], for cell separation, focusing, isolation, analysis, and diagnosis;
- others [20].

Within these techniques, the most suitable for bioparticle manipulation are the methods based on electrical fields as a result of their strong controllability, high efficiency, and easy operation [21]. One such electrokinetic phenomena which allows the analysis of the preferential formation and growth of biofilms within microfluidic devices is dielectrophoresis (DEP), which relies on the control of electrically neutral particles. When suspending such particles or cells in a non-uniform electric field, they will be polarized into dipoles, and the net force acting upon them will drive the particles or cells towards a high field gradient region (positive DEP) or push away from it (negative DEP) [21,22].

Bacterial biofilms are structured communities of bacterial cells surrounded by an extracellular (polymeric) matrix (ECM) that firmly attach to biotic or abiotic surfaces [23]. The aggregation, or cell–cell interaction, can occur both in surface-attached biofilms, where one layer of the multi-layered biofilm interacts with a substratum, or in flocs, otherwise considered mobile biofilms. Either way, there is a clear distinction in their behaviour, compared to free-living bacterial cells, as a result of the social and physical interactions occurring between the cells and the ECM [24]. The ECM represents a three-dimensional scaffold-like structure responsible for the adhesion of the biofilm to surfaces, providing mechanical stability and complex microenvironments essential for the biofilm lifestyle [25,26]. It is a predominantly aqueous medium containing structural and functional components [24]

both soluble, such as gel-forming polysaccharides [25] and proteins, as well as insoluble, such as amyloids [27], cellulose [28], fimbriae, pili and flagella [27]. Besides the various proteins from the extracellular matrix (ECM), there are also proteoglycans and glycoproteins that are secreted locally and assembled into a spatially organized architecture in close association with the surface of the cells that produced them [29].

Each of the biofilm formation and growth stages is influenced and governed by different environmental factors. For biofilms growing on a substratum, the first step in the formation process is the adherence of the planktonic microorganisms to a surface, which is strongly determined by the surface quality of the substratum [30], and the existence of surface proteins such as SadB or LapA on the substratum [31]. This stage is followed by the development of microcolonies, the secretion of the extracellular polymeric matrix, and the development of a three-dimensional biofilm community, phases influenced by aerobic/anaerobic environmental conditions [32], temperature and shear stress [33]. The last step is the detachment of the microorganisms and their dissemination into the environment, the stage dependent on cell motility and ECM degradation, as well as on physical factors, such as the shear force [31,34,35].

An example of Gram-positive cocci involved in a wide variety of human infections and known for its ability to successfully develop biofilms on both biotic and abiotic surfaces [36,37] includes Staphylococcus aureus. Usually, Staphylococcus aureus produces a complex biofilm structure with multiple layers of glycocalyx [37]. In addition, the Staphylococcus aureus extracellular matrix has a fibrous structure [29], similar to that of fibrils/microfibrils structure type, containing nanocellulose fibre from biofilm of *Komagataeibacter hansenii* [30]. The adhesion and maturation of biofilm are usually correlated with the production of an antigen responsible for adhesion, known as polysaccharide intercellular antigen (PIA). The synthesis of PIA starts from UDP-N-acetylglucosamine and is mediated by the products of the intercellular adhesion locus [38]. The interaction of these with different electric potentials has not yet been studied to our knowledge. Thus, this study provides the first insight into the interaction between the first steps in biofilm formation, adhesion, and formations of microcolonies, at different electric potentials.

A growing body of literature has investigated the possibility of using microfluidic devices for the study of bacterial microorganisms, such as Vibrio parahaemolyticus in glass devices [39], E. coli in PDMS, PDMS/PMMA [40], and PMMA [41] microfluidic devices, Cordyceps militaris in hydrogel/PMMA devices [42], E.coli, Salmonella Typhimurium, Listeria monocytogenes in paper-based devices [43], and Pseudomonas aeruginosa and Staphylococcus aureus in nitrocellulose membrane-based devices [44,45]. However, few researchers have addressed the issue of studying highly virulent and antibiotic-resistant bacterial pathogens, such as *Enterococcus faecium*, *Staphylococcus aureus*, *Klebsiella pneumoniae*, *Acinetobacter baumannii*, *Pseudomonas aeruginosa*, and *Enterobacter spp.*, through the means of 3D printed microfluidic devices.

Previously we have demonstrated that it is possible to influence the preferential formation and growth of bacterial biofilms by the application of a non-uniform electric field within a microfluidic device [46]. However, it is essential to understand to what degree the DEP forces affect the preferential formation of biofilms and how the structure of bacterial biofilms is affected in in situ-like environments when subjected to high DC electric potential differences.

The aim of this study is to describe the effect of the different electric potentials upon the preferential formation and growth of *Staphylococcus aureus* ATCC 25923 biofilms in 3D printed microfluidic devices.

## 2. Materials and Methods

### 2.1. Device Setup

The microfluidic devices were fabricated using a Crealty3D Ender 5 printer (Shenzhen Creality 3D Technology CO., Ltd., Shenzhen, China) and a PLA filament (VerbatimTM—Mitsubishi Kagaku Media Co., Ltd., Tokyo, Japan), based on a design described in previous

work (Figure 1) [46]. The initial model was created using the SolidWorks 3D CAD Software (Education Edition 2019–2020, Dassault Systèmes, Vélizy-Villacoublay, France), while the translation to a G-code was done using the UltiMaker Cura 4.13 free 3D printing software. One side of the device accommodated a system perpendicular to the main channel, incorporating 9 copper electrodes, each with the diameter of Ø 1 mm (side B). The other side allowed the insertion of an electrode, with a diameter of Ø 1 mm, parallel to the main channel (side A) (Figure 2).

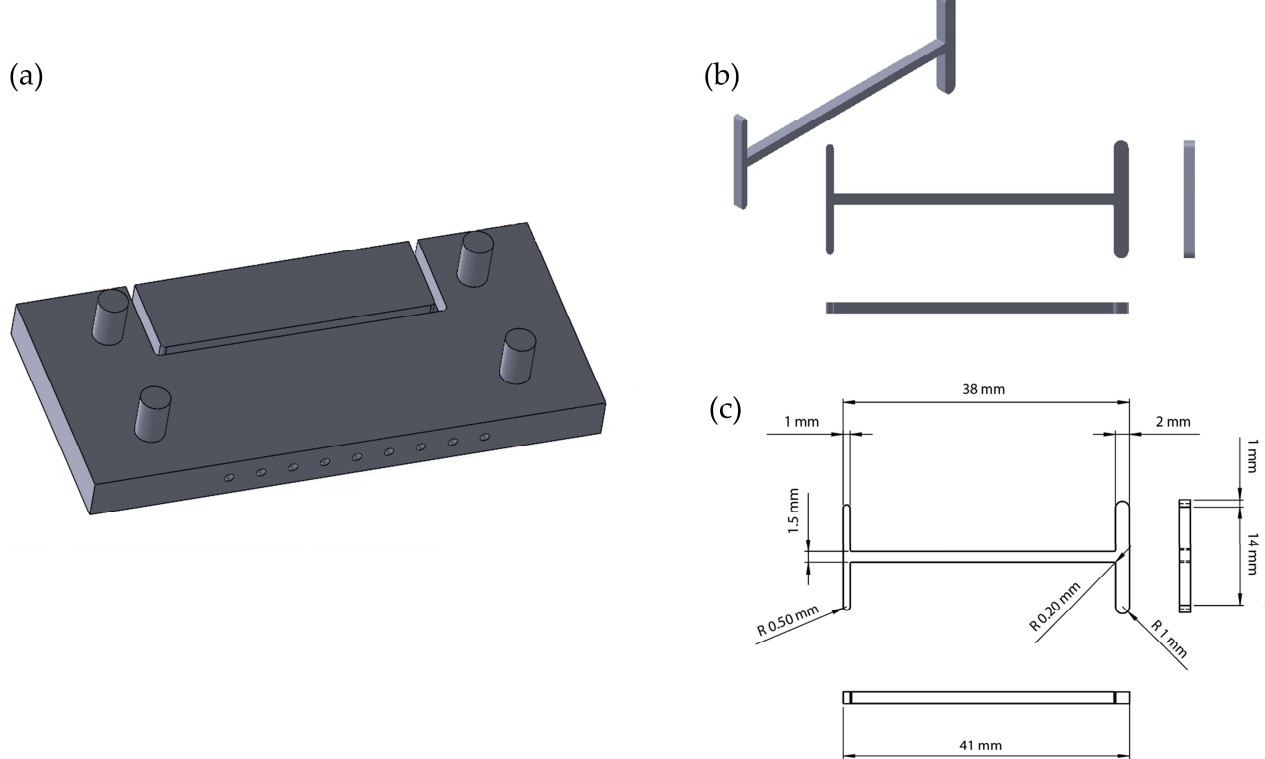

**Figure 1.** Graphical representation of the (**a**) Microfluidic device; (**b**) Microfluidic channel; (**c**) Overall microfluidic channel dimensions.

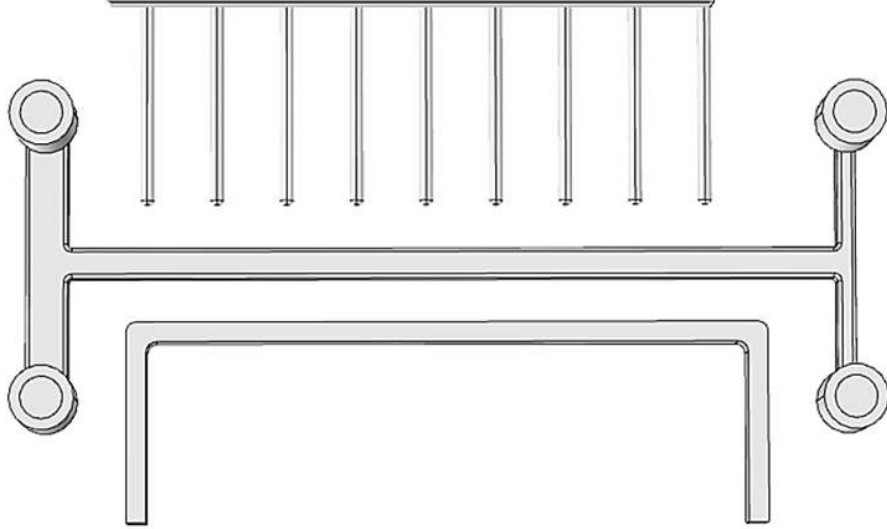

**Figure 2.** Graphical representation of the placement of the electrodes along the main channel.

Using crocodile clamps connected to an Aim-TTi QPX600DP Bench Power Supply (600 W, 2 Output, 0 → 80 V, 0 → 50 A), the parallel electrode was set at ground potential,

while the perpendicular electrode system was set to values varying from 10 to 60 V. This setup allowed the generation of a non-uniform electric field across the main channel.

Using Ø 1 mm diameter microfluidic tubes, the systems' inlets were connected to two syringes: one filled with 3 mL bacterial suspension ($7.5 \times 10^8$ CFU/mL); and the other one filled with 3 mL nutrient broth (MBH, Bio-Rad, Marnes-la-Coquette, France), to ensure a proper growth medium for the bacterial biofilms.

### 2.2. Biofilm Cultivation, Formation, and Growth

Choosing a relevant bacterial strain for this study was a challenging step. One of the most common bacteria involved in biofilm-related infections is the Gram-positive *Staphylococcus aureus* ATCC 25923. The bacterial strain was cultivated on 5% sheep-blood agar (bioMerieux, 107 Marcyl' Étoile, France) plates and incubated for 24 h at 37 °C. By picking up colonies from the agar plates and mixing them in 5 mL saline solution, 2.5 McF bacterial suspensions were prepared.

The obtained solutions and suspensions were drawn in syringes and pushed through the microfluidic systems using a SP230iwZ Syringe Pump (WPI), at a flow rate of 0.3 mL/min, at room temperature (Figure 3). Once the main channel started filling up with fluid, the Power Supply Bench was turned on, and several potential values, varying between 10 and 60 V were applied to the electrodes. This potential was applied until the flow of fluids was stopped within the microfluidic devices. The next step consisted of cultivating the microfluidic devices at 37 °C for 120 h, followed by an inactivation step using UV light. Previous to the analysis, the devices were cut along the main channel using a Robotec laser cutting machine, and for the scanning electron microscopy (SEM) analysis, the samples were washed with 3 mL absolute alcohol solution and air dried.

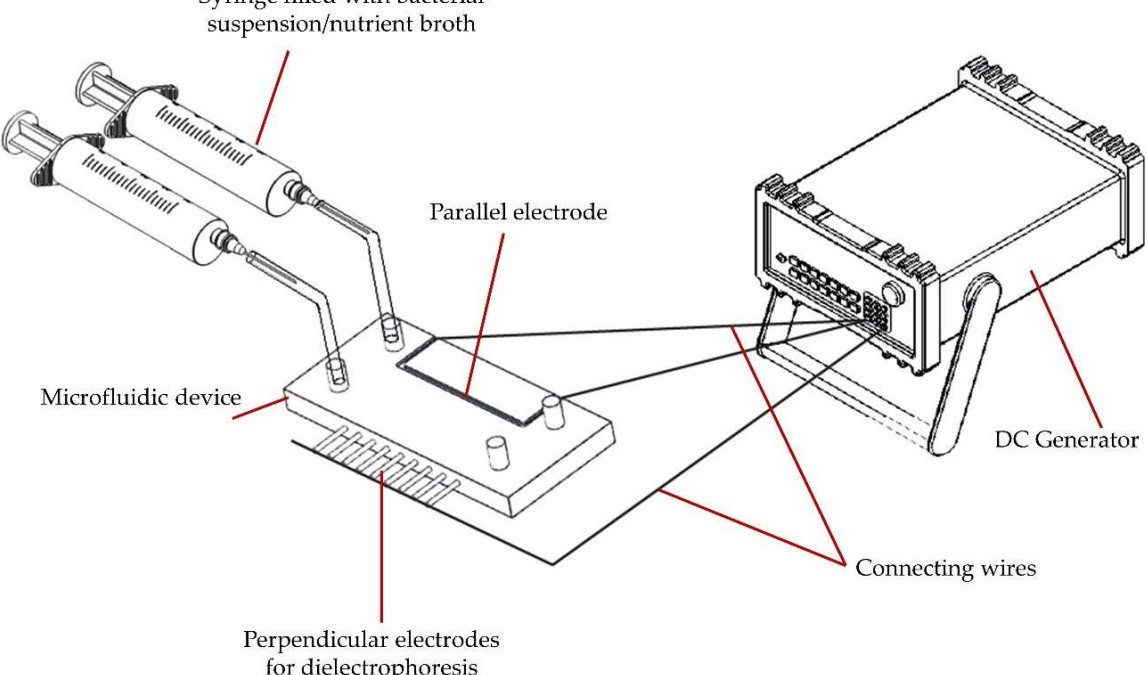

**Figure 3.** Schematic representation of the device setup, including a syringe pump, the syringes filled with bacterial suspension and nutrient broth, the microfluidic device with perpendicular and parallel electrodes, and the DC generator.

### 2.3. Analysis Methods

For the assessment of biofilm formation and growth, both quantitative and qualitative methods were used. To quantify the amount of biofilm formed inside the microfluidic devices, an adaptation of the crystal violet technique was used [47]. The cut sides of the

systems (marked with A for the side accommodating the parallel electrode and B for the side including the perpendicular electrodes) were first washed with 5 mL saline solution to remove potential residual planktonic cells, followed by staining with 1% crystal violet, for 2 min. Further, the systems were washed 3 times using alcohol to remove the dye, and the final washing solution was collected and stored in transparent tubes. This solution was diluted 1:10 with ethanol-acetone and used for UV-Vis analysis at 590 nm ($\lambda_{max}$ for crystal violet) (Figure 4).

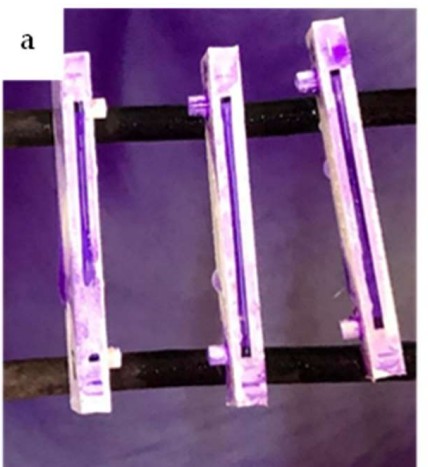 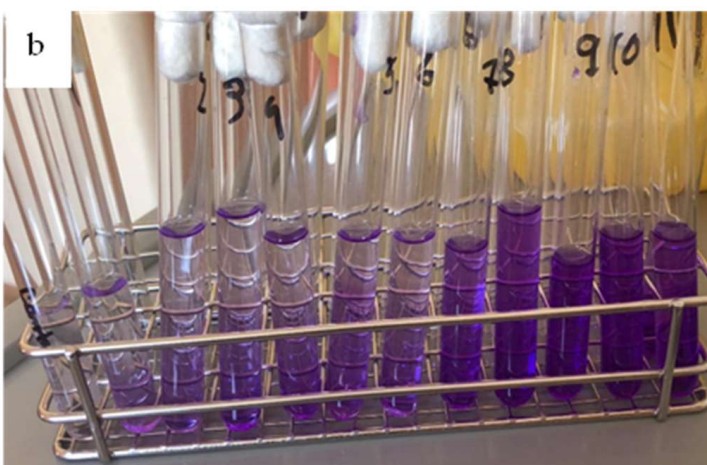

**Figure 4.** (**a**) Cut side of the microfluidic systems after the crystal violet staining; (**b**) Diluted solutions for UV-Vis investigation.

The images of the biofilms' surface topography were recorded using a scanning electron microscope (SEM) Hitachi CFE SU8230 (Hitachi High-Tech Corporation, Tokyo, Japan) (CFE = cold field emission) operating at an accelerating voltage of 10 kV and magnifications of $50\times–10,000\times$. All samples were coated before viewing with a 5-nm conductive gold layer to increase the contrast level in the Secondary Electron Images (SEI).

## 3. Results and Discussions

### 3.1. Distribution of the Electric Field

Using the COMSOL Multiphysics software (version 4.3, COMSOL Inc., Stockholm, Sweden), a simulation of the electric field distribution inside the microfluidic devices was performed. For the parallel electrode, the potential was considered ground potential, while the perpendicular electrodes were set to values varying between 10 and 60 V. The remaining components of the microfluidic device were set to insulating boundary, whereas the fluid flowing inside the microfluidic channel was considered to have the same electric properties as water. Figure 5 shows the electric field distribution (V/m) inside the microfluidic devices, for each electric potential difference, with a focus on the electrodes and the central channel. As can be observed, the configuration of the perpendicular electrodes allows for the concentration of the electric field in points, enabling higher gradient regions to form along the main channel. Another noticeable aspect is the high gradient region that forms around the first and last electrode disposed perpendicularly to the main channel. As the electric potential difference increases, the distance between these high-gradient regions decreases, leading to an almost uniform electric field at around a potential of 60 V (Figure 5g). Considering the positive dielectrophoretic behaviour displayed by the *Staphylococcus aureus* ATCC 25923 cells [18] and the decrease in the distance between the high electric gradient regions could lead to the deposition of a more uniform bacterial biofilm along the side B (inlet side for the bacterial suspensions).

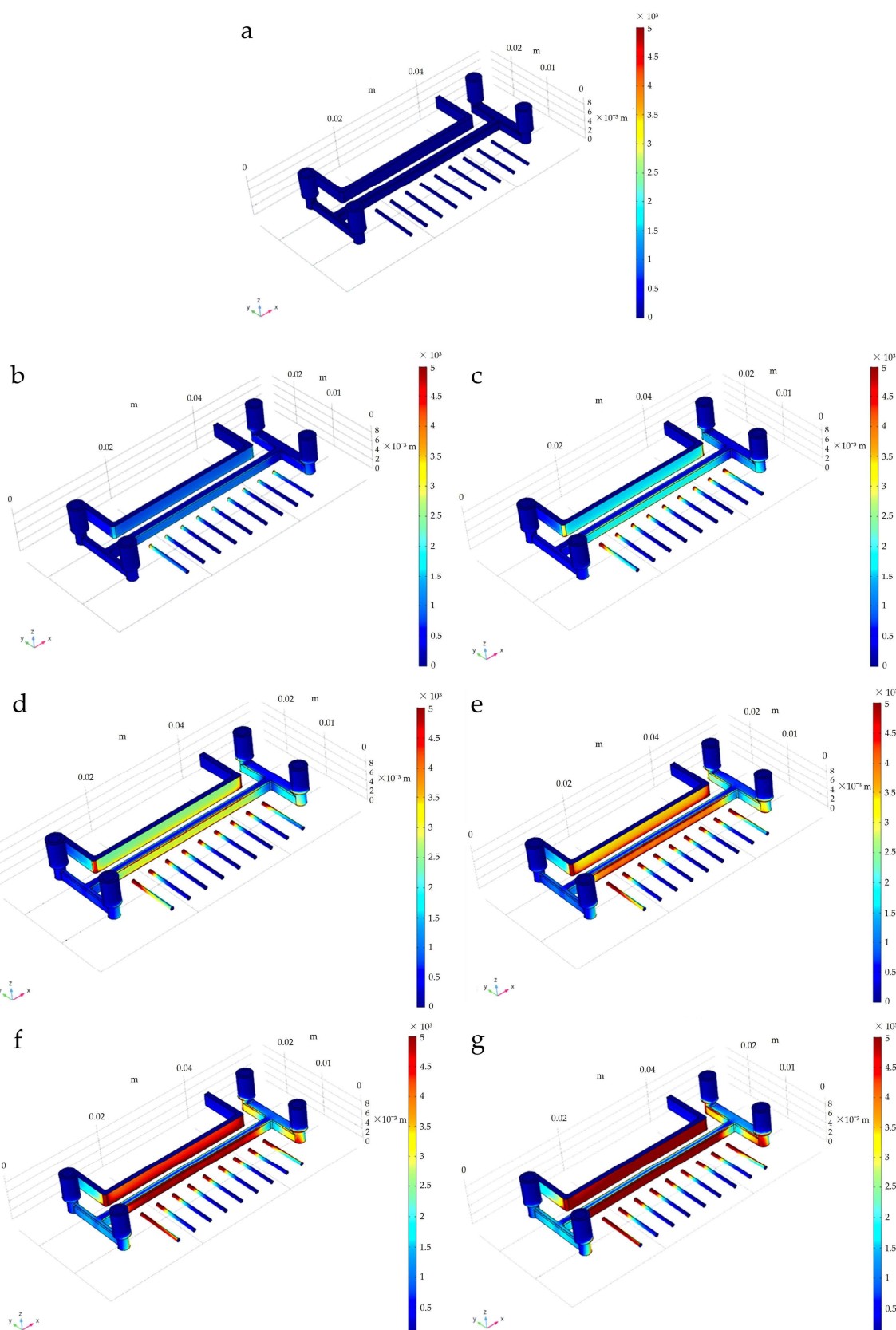

**Figure 5.** Electric field distribution (V/m) inside the microfluidic channels at (**a**) 0 V; (**b**) 10 V; (**c**) 20 V; (**d**) 30 V; (**e**) 40 V; (**f**) 50 V; and (**g**) 60 V.

### 3.2. Scanning Electron Microscopy (SEM)

The SEM images (Figure 6) describe, at ×100 and ×10,000 magnifications, the morphology and topography of the surface of the biofilms formed inside of the microfluidic devices at voltages ranging from 10 V to 60 V. Figure 6(a1,a3) illustrate the structure of the biofilm formed at 10 V voltage on sides A and B, respectively, at a magnification of ×100. As can be observed, the filament deposition lines, resulting during the printing process, are more defined on the A side of the channel, suggesting a thinner layer of the deposited biofilm compared to that on side B. Another aspect to be taken into consideration is the more grainy structure of the biofilm on side B, with more systematically organized bacterial colonies. At a closer look at the extracellular matrix, at a magnification of ×10,000, Figure 6(a2,a4) display a denser structure of the biofilm on side B.

A similar situation occurs in the case of the samples prepared at 20 V, and 30 V. As shown at ×100 magnification in Figure 6(b1,b3,c1,c3), a thicker biofilm can be observed on side B, compared to side A. Figure 6(b4,c4) allow the visualization of the cell clusters within the ECM, suggesting the existence of organized structures of the cells under and in between the pores and layers of the protective ECM. Figure 6(c2) indicates the existence of only the ECM inside the channel, with no visible individual cells. This could be explained by both the flow rate acting upon the cell attachment inside the channel as well as by choice of area for microscopic visualization (images taken closer to the central area of the microchannel).

A common behaviour is observed on side B of the samples prepared at 40 V, 50 V, and 60 V, where the bacterial biofilm layer appears to cover the whole width of the channel, with the deposition lines completely disappearing under the biofilm (Figure 6(d3,e3,f3)). In Figure 6(d1), it is noticeable that for the 40 V sample, on side A, the deposition lines are still visible, with noticeable gaps between the filaments, whereas for the 50 V sample's side A (Figure 6(e1)), the deposition of the biofilm seems more "hill" or "dome" like. For the 60 V sample, the filament deposition lines disappear completely on side A (Figure 6(f1)) as well as on side B, suggesting a change in behaviour due to the more uniform electric field.

Figure 6(e2,e4) illustrate whole-structured cells, whereas Figure 6(f2,f4) present a more fractured or damaged structure, with cells less round and organized. This phenomenon could be a result of the high electric gradient interacting and destroying the cell walls.

According to the SEM images, the dielectrophoretic effect of the non-uniform electric field upon the bacterial cells allows for the formation of a more robust, thicker, and overall more compact biofilm on the B side of the channel that accommodates the perpendicular electrodes, therefore conferring the cells a positive dielectrophoretic behaviour. Differences in the biofilms' overall structures are more noticeable after the 40 V mark when the dielectrophoretic effect seems to intensify, the biofilms becoming more dense, all together with a more comprehensive structure.

The effects of the dielectrophoretic forces seem to decrease once the 60 V mark is reached, as suggested both by the simulations and the SEM images. Cell structures seem to lose definition, while the biofilm takes the shape of a comprehensive mass rather than intercommunicating colonies (Figure 6(f1,f3)).

Another noticeable aspect emphasized by the SEM images is the close cell–cell contact occurring in the biofilm and the fact that the biofilm matrix may act to stabilize contacts between neighboring bacteria [48,49]. The biofilm's architecture appears as an extended, tightly packed biomass in which cells cluster in microcolonies with relatively uniform adhesion [50]. At a careful examination at ×10,000 magnification, it can be observed, in both the bottom and upper regions of the ECM, the presence of relatively longitudinal aligned filaments, resembling the fibre-like structure of ECM of *S. aureu* biofilms, that might facilitate its adherence to catheters and foreign material as biofilms [51]. In addition, the pores and the channel between microcolonies observed in the ECM of the biofilm could be considered as belonging to the "rudimentary circulation system" that facilitates the nutrient transport [52] inside the biofilm [53]. In the case of Staphylococcus aureus biofilms, the phenol-soluble modulin surfactant peptides play a key role both in the formation of

these channels within the biofilm and also in surface modification and colony spread, respectively [54].

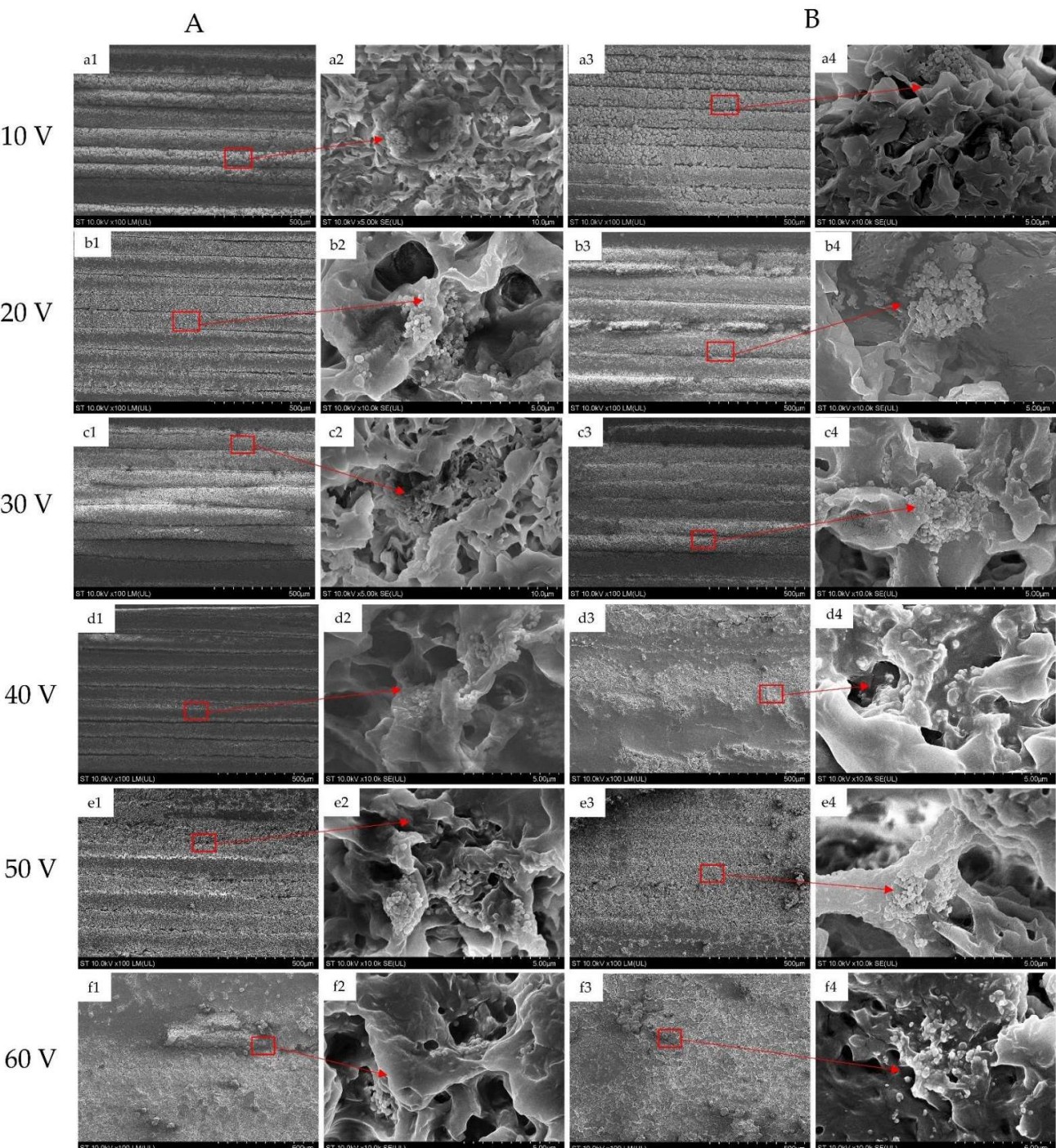

**Figure 6.** SEM images of the bacterial biofilms alongside (**A**) at 10 V ((**a1**)—×100; (**a2**)—×10,000); 20 V ((**b1**)—×100; (**b2**)—×10,000); 30 V ((**c1**)—×10; (**c2**)—×10,000); 40 V ((**d1**)—×100; (**d2**)—×10,000); 50 V ((**e1**)—×100; (**e2**)—×10,000); and 60 V ((**f1**)—×100; (**f2**)—×10,000); and side (**B**) at 10 V ((**a3**)—×100; (**a4**)—×10,000); 20 V ((**b3**)—×100; (**b4**)—×10,000); 30 V ((**c3**)—×100; (**c4**)—×10,000); 40 V ((**d3**)—×100; (**d4**)—×10,000); 50 V ((**e3**)—×100; (**e4**)—×10,000); and 60 V ((**f3**)—×100, (**f4**)—×10,000). The rectangular areas and the arrows correspond to the observation area in the ×100 magnification images.

### 3.3. Quantitative Analysis

Figure 7 shows the graphical representation of the biofilm concentration between sides A and B (parallel and perpendicular electrodes) after a statistical t-distribution analysis. For each electric potential, three different samples were prepared for the quantitiative analysis, the results representing an average of the measurements carried out on these samples. As can be observed, the overall trend of the graphic suggests the formation of a thicker biofilm on the side of the channel accommodating the perpendicular electrodes, confirming the previously stated positive dielectric properties of the bacterial cells. However, there is a slight variation at the 40 V mark, where the concentration of the biofilm is greater on the B side of the channel. This behaviour needs further investigation and tests.

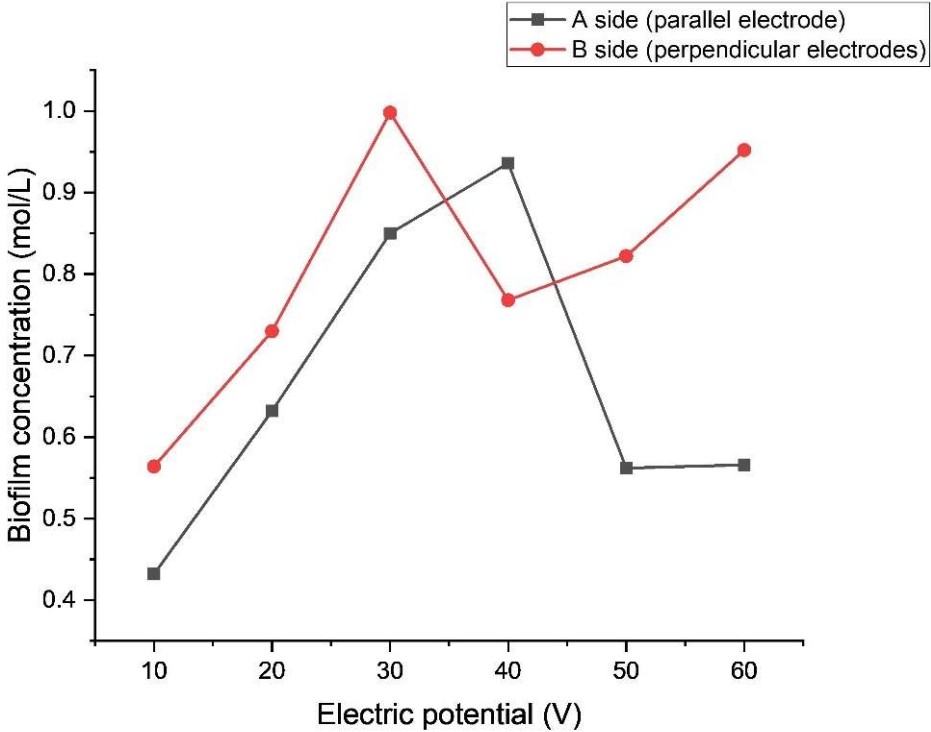

**Figure 7.** Graphical representation of the biofilm concentration inside the A and B parts of the channel as a function of the applied electric potential difference.

Another noticeable aspect is that above the electric potential of 40 V, the *p*-value is less than 0.05, indicating a statistically significant difference between the concentrations of biofilms along the two sides of the channel. This is a good indication of the fact that up to that electric potential value, the dielectrophoretic forces' effect on the cells, however existent, is not as impactful as those above this value. Above 40 V, the effect of the dielectrophoretic forces increases, creating a greater difference in biofilm concentration between the two sides. For the side of the channel accommodating the parallel electrode above 50 V, the concentration of the biofilm seems to remain stationary, while the values of the concentration for the B side seem to decrease. This could be an indication of the decreasing distance between the high electric gradient regions along the main channel, as seen in the simulations.

Certainly, this work has its limitation, one of which is the small size of the samples, with only five samples used for the statistical analysis. Another limitation is given by the timeframe used for incubation. As the amount of biofilm was determined only after the 120 h mark, future research should focus on the influence of the cultivation period on the quantity of biofilm formed inside the microfluidic devices.

Several limitations are given by the scale of the device and the placement of the electrodes along the central channel. Future work should explore the effects of increasing



the length of the microfluidic device, as well as the different possibilities of positioning the parallel electrodes along the main channel in order to alter the distribution of the electric field.

Another direction for future research could include analysis of the effect of the different electric potentials (in magnitude and uniformity) on the biofilm attachment and formation for several other bacterial types inside 3D printed microfluidic devices.

## 4. Conclusions

This paper analyses the influence of dielectrophoretic forces at different electric potentials on the preferential formation and growth of *Staphylococcus aureus* ATTC 25923 biofilms in 3D printed microfluidic devices. According to the simulations, the SEM images, and the quantitative analysis, there is an increase in the dielectrophoretic effect on the *Staphylococcus aureus* ATCC 25923 cells between the electric potentials of 40 and 50 V and a decrease in their impact around the value of 60 V. Around an electric potential of 60 V, the electric field distribution becomes almost uniform along the main channel, allowing for a switch from dielectrophoretic forces to electrophoretic forces acting upon the bacterial cells.

The present findings can be used for future research referring to the preferential formation and growth of different types of biofilms in in situ-like conditions.

**Author Contributions:** Conceptualization, A.C. (Alexandra Csapai) and C.O.P.; methodology, A.C. (Alexandra Csaoai), C.O.P. and D.A.T.; validation, C.O.P., C.C. and V.P.; formal analysis, S.T., A.C. (Alexandra Ciorîţă), N.T., B.M. and V.P.; investigation, S.T. and N.T.; resources, C.O.P. and C.C.; data curation, R.M.M., C.O.P., N.T., B.M. and V.P.; writing—original draft preparation, A.C. (Alexandra Csapai) and D.A.T.; writing—review and editing, C.O.P., V.P. and C.C.; visualization, A.C. (Alexandra Csapai) and C.O.P.; supervision, C.O.P. and C.C.; project administration, C.O.P. and C.C.; funding acquisition, C.O.P. and C.C. All authors have read and agreed to the published version of the manuscript.

**Funding:** This research received no external funding.

**Institutional Review Board Statement:** Not applicable.

**Informed Consent Statement:** Not applicable.

**Data Availability Statement:** Not applicable.

**Acknowledgments:** N.T. acknowledges support by a grant of the Romanian Ministry of Education and Research, CNCS-UEFISCDI, project number PN-III-P4-ID-PCE-2020-1607, within PNCDI III.

**Conflicts of Interest:** The authors declare no conflict of interest.

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
