# Peer review of "Study of the Influence of the Dielectrophoretic Force on the Preferential Growth of Bacterial Biofilms in 3D Printed Microfluidic Devices"

_applsci, doi:10.3390/app13010060_

Round 1
Reviewer 1 Report
The authors investigated “Study of the Influence of the Dielectrophoretic Force on the Preferential Growth of Bacterial Biofilms in 3D Printed Micro-fluidic Devices”. The manuscript falls into the scope of the journal. The manuscript is well-written and contains valuable information. The structure of the MS is considerably organized. The authors present adequate information in the methodology to allow the reproduction of the estimated methods properly. However, introduction and discussion section need improvement.
1. Briefly explain biofilm and its formation process, abiotic and biotic surfaces used in biofilm formation assays, biofilm (EPS) matrix components, factors (nutritional, environmental and genetic factors) affecting biofilm formation, and both beneficial and detrimental effects of biofilms in the introduction section with recent references
e.g.,
[1] https://doi.org/10.1016/j.envpol.2022.12023
[2] https://doi.org/10.1016/j.tim.2020.03.016
[3] https://doi.org/10.1038/s41598-022-09519-9
[4] https://doi.org/10.1073/pnas.2123469119
[5] https://doi.org/10.1038/nrmicro2415
2. Several recent studies (SEM analysis) shown that bacterial biofilms are compact, highly fibrous and contain numerous nanowires and nanocellulose bundles e.g.,
[1] https://doi.org/10.1016/j.envpol.2022.12023
[2] https://doi.org/10.1016/j.tim.2020.03.016
[3] https://doi.org/10.1038/s41598-022-09519-9
[4] https://doi.org/10.1073/pnas.2123469119
[5] https://doi.org/10.1038/nrmicro2415
It would be better if authors discuss this statement in the results and discussion section.
3. Write conclusion within 100 words.
4. Line # 132: Specify bacterial colony forming unit.
5. Line # 138: Briefly write why 5% sheep-blood agar is used to culture the test bacterium.
6. Line # 159: A citation is needed for CV method.
7. Line # 162: Correct as “with 1% crystal violet” instead of with crystal violet 1%
8. Change ml to mL throughout the manuscript.
9. Change 37ºC to 37 ºC throughout the manuscript
10. Change 120h to 120 h
11. Line # 248: Correct accomodation to accommodation

Author Response
Thank you for giving us the opportunity to submit a revised draft of the manuscript: “Study of the Influence of the Dielectrophoretic Force on the Preferential Growth of Bacterial Biofilms in 3D Printed Microfluidic Devices” for publication in the Journal Applied Sciences. We appreciate the time and effort that you dedicated to providing feedback on our manuscript and are grateful for the insightful comments on and valuable improvements to our paper. We have incorporated most of the suggestions made by the reviewer. Those changes are highlighted within the manuscript.

Reviewer 2 Report
The comments to the authors is attached separately.

Author Response

(The authors gave the same response as above.)

Round 2
Reviewer 2 Report
Dear Editor,
The authors have responded all the queries raised during the review process. Hence, I would like to recommend you to accept the manuscript as it is.
Thank you
D. Rajamani